# Psychometric Testing of the Bahasa Version of the Brief Illness Perception Questionnaire among Indonesians with Type 2 Diabetes Mellitus

**DOI:** 10.3390/ijerph18189601

**Published:** 2021-09-12

**Authors:** Yohanes Andy Rias, Andi Hayyun Abiddin, Nuh Huda, Sri Handayani, Healthy Seventina Sirait, Li-Chung Pien, Shuen-Fu Weng, Hsiu-Ting Tsai

**Affiliations:** 1Faculty of Health and Medicine, Institut Ilmu Kesehatan Bhakti Wiyata Kediri, College of Nursing, Kediri 64114, Indonesia; yohanes.andi@iik.ac.id; 2Post-Baccalaureate Program in Nursing, College of Nursing, Taipei Medical University, Taipei 11031, Taiwan; andy5240@tmu.edu.tw; 3Nursing Department, College of Nursing, Politeknik Kesehatan Kemenkes Malang, Malang 65112, Indonesia; andi_hayyun@poltekkes-malang.ac.id; 4Sekolah Tinggi Ilmu Kesehatan Hang Tuah, Surabaya 60244, Indonesia; nuhhuda@stikeshangtuah-sby.ac.id; 5Faculty of Nursing, Universitas Airlangga Surabaya, Surabaya 60132, Indonesia; 6Nursing Program, Sekolah Tinggi Ilmu Kesehatan Yogyakarta, Yogyakarta 55162, Indonesia; handayanis359@gmail.com; 7Nursing Program, Sekolah Tinggi Ilmu Kesehatan Cirebon, Cirebon 45153, Indonesia; healthy.seventinasirait@gmail.com; 8Department of Internal Medicine, Division of Endocrinology and Metabolism, Taipei Medical University Hospital, Taipei 11031, Taiwan; sfweng@ntu.edu.tw; 9Department of Internal Medicine, Division of Endocrinology and Metabolism, School of Medicine, College of Medicine, Taipei Medical University, Taipei 11031, Taiwan

**Keywords:** Bahasa version, brief illness perception questionnaire, psychometrics, reliability, type 2 diabetes, validity

## Abstract

The Brief Illness Perception Questionnaire (B-IPQ) has been recommended to validate illness perception. Nevertheless, this measurement has yet to be validated with an assessment of the construct and convergent validities and reliability in Indonesia. Our study aimed to psychometrically test the 8-item B-IPQ among Indonesians with type 2 diabetes mellitus (T2DM). Data included 294 patients with T2DM with stratified multistage clustering. The 36-item Short Form Survey, 21-item Depression Anxiety Stress Scale, and fasting blood glucose (FBG) were used to examine convergence and divergence. The validity analysis included the construct and convergent validities with significant person correlations. Cronbach’s alpha, composite reliability (CR), and average variance extracted (AVE) were used to assess reliability. Confirmatory and exploratory factor analyses indicated a multidimensional structure, including cognitive with a five-item structure and emotional illness representation with a three-item structure, with an acceptable goodness of model fit. The tool revealed good internal consistency for the cognitive, emotional, and overall domains and was positively moderately correlated with FBG, stress, anxiety, and depression but negatively correlated with the overall quality of life and mental and physical component scores. Findings provide empirical evidence that the Bahasa version of the B-IPQ showed adequate internal consistency, exploratory and confirmatory, and thus is valid and reliable for illness perception assessments among Indonesians with T2DM.

## 1. Introduction

Type 2 diabetes mellitus (T2DM) is a category of complex metabolic disorders caused by an elevated level of glucagon secretion in the presence of insulin resistance (IR) [1]. This disease has now reached a global epidemic level and has escalated mortality risks [2,3]. The International Diabetes Federation has predicted that globally the prevalence of diabetes will exceed 578 million by 2030 and rise to 700 million by 2045. This challenge also exists in Indonesia, where the incidence of diabetes is projected to grow from 10.7 million in 2019 to 16.6 million in 2045 [2]. Notably, it was reported that IR is the major cause of low illness perception levels of T2DM [4]. Illness perceptions refer to personal beliefs and their impacts on health behaviors [5,6]. Consequently, these conditions may impair self-management and promote the development of possible complications [7]. It is critical to have an accurate and appropriate instrument for the early measuring of illness perceptions among the Indonesian population with T2DM.

The original individual Illness Perception Questionnaire (IPQ) was used to measure cognitive elements with five scales that assessed identity, cause, timeline, consequences, and cure control when facing an illness, and it was developed by Weinman et al. [8]. Moreover, a revised version of this scale, the Illness Perception Questionnaire-Revised (IPQ-R), enlarged the original scale by providing over 80 items, separating the control elements into personal and treatment control, and integrating a cyclical timeline, an overall comprehension of illness factors, and an emotional subscale [9,10], but in certain cases, a lengthy questionnaire will be insufficient [10], requires a greater response burden, and results in nonresponses and attrition [11]. Remarkably, the English version of the Brief IPQ (B-IPQ) includes nine items, covering three aspects: cognitive dimensions of illness representations (cognitive representation), emotional dimensions of illness representations (emotional representation), and illness comprehensibility [10,12]. These nine items capture the conceptual breadth of illness perceptions.

A systematic review and meta-analysis found that the concept of the B-IPQ had been adopted and validated in 36 countries such as Australia, Colombia, New Zealand, Germany, Portugal, China, Taiwan, Singapore, and Japan [12]. The original English and translated versions of the B-IPQ had been validated in patients with various chronic illnesses, including patients with diabetes [12,13,14]. Interestingly, in 2005 in New Zealand; the UK; and Texas, USA, the B-IPQ version was completed by patients with renal disease, myocardial infarction, asthma, allergies, and T2DM for the first time [10].

Previous research in Poland illustrated the good internal consistency of the B-IPQ as satisfactory when assessing participants with myocardial infarction, neurological diseases, rheumatoid arthritis, and lupus [15]. Moreover, the B-IPQ was psychometrically tested in Ghana among T2DM patients [16] and was established in Korea with tuberculosis patients [17]. However, the validity and reliability of the B-IPQ have never been rigorously estimated for application in Indonesians with T2DM. Thus, psychometric assessments of the B-IPQ in other multicultural countries, particularly among Indonesians with T2DM, are empirically required. Moreover, there are no evidential studies related to how to measure B-IPQ scores in terms of whether they should be represented as eight items of illness on two subscales (cognitive and emotional components of representations) among Indonesians with T2DM.

In addition to standard psychometric considerations such as reliability and validity, satisfying the demands of construct validity is a concern and challenge, particularly in Indonesia with T2DM patients’ perceptions to understand their illness and escalate the quality of care they receive related to their illness. According to previous research on depression, stress, anxiety, and blood glucose, it was proven that the B-IPQ was easy to use and had satisfactory convergent validity [11,13]. Moreover, the construct and convergent validities need to be further assessed in various populations [18,19]. Therefore, this psychometric test of the B-IPQ focused on patients with T2DM in Indonesia with an assessment of the construct and convergent validities and reliability.

Therefore, to address this knowledge gap, the study aimed to examine the psychometric properties of the Bahasa version of the B-IPQ in Indonesians with T2DM. Specifically, we explored the data fit for the eight items with a two-factor loading. We also evaluated the internal consistency and the construct and convergent validities of the B-IPQ Bahasa version.

## 2. Materials and Methods

### 2.1. Design

A cross-sectional study with community-based stratified multistage clustering was implemented between 1 July and 30 November 2018, and an exploratory factor analysis (EFA) with confirmatory factor analysis (CFA) was conducted in East Java, a province in Indonesia. The translation process and psychometric analytical procedures of the Bahasa version of the B-IPQ were explored.

### 2.2. Participants

For the first phase of the community-based stratified multistage cluster, we conveniently selected East Java province and divided it into 38 regions. For the second phase, two rural and two urban areas were chosen from those 38 regions. In the final phase, eight community clinics were randomly chosen from the four regions for data collection; three of the selected community clinics declined our invitation to participate in the research. In the end, five community-based area clinics participated in our work for respondent recruitment. The inclusion criteria for T2DM were (1) an Indonesian national 18 to 79 years old, (2) having agreed to participate and completed all assessments, and (3) with a diagnosis of T2DM validated by a medical doctor according to American Diabetes Association (ADA) criteria [1]. Participants who (1) had a score of ≤24 on the mini-mental state exam, (2) were pregnant, (3) had a disability, (4) used antidepressants, or (5) had auditory deficiencies were excluded from the study. Considering the sample size [20], at least 240 individuals with T2DM were required for the goodness of fit index and an adequate and interpretable value of the B-IPQ. Considering the potential attrition rate or missing data of 20%, we increased our total sample size to approximately 300. However, six participants did not fill out all assessments. Consequently, the sample size was 147 participants for EFA and 147 participants for CFA, and 294 individuals with T2DM were included

### 2.3. Procedure

The Bahasa Indonesia version of the B-IPQ was assessed for translation process and psychometric analytical procedures including the following: (1) We translated the English version of the B-IPQ into Bahasa Indonesia by a professional bilingual translator following recommended guidelines for cross-cultural adaptation assessment [21]. In brief, Indonesian-speaking academics were first contacted to review the translated version for grammatical accuracy and clarity. Thus, four independent bilingual translators completed the back translation of the Bahasa edition into English. Moreover, the final Bahasa version was obtained after comparing the original and back-translated questionnaires. Translators were instructed to use simple sentences and avoid metaphors, colloquial terminology, and hypothetical statements. (2) A pilot test of 10 Indonesians with T2DM was evaluated to clarify the wording and ease of understanding of the final version. (3) We reviewed cognitive debriefing results and the finalized version (content validity index (CVI) for the total scale = 0.95 with four expert reviews). (4) Finally, we conducted a psychometric analysis of the internal consistency and construct, convergent, and divergent validities; the analytical process and results are summarized in Table 1.

### 2.4. Instruments

A survey using a structured questionnaire was utilized to collect data regarding demographics, including age, gender, marital status, income, education, smoking status, and diabetes duration. The questionnaire was confirmed to be accurate with validity and reliability in previous studies [26,27]. 

#### 2.4.1. Assessments of Illness Perception 

Illness perceptions were assessed using the B-IPQ. The original English version of the B-IPQ includes nine items, covering three aspects: cognitive dimensions of illness representations (cognitive representation), emotional dimensions of illness representations (emotional representation), and illness comprehensibility. Cognitive representation is assessed by five items addressing consequences (item 1), timeline (item 2), personal control (item 3), treatment control (item 4), and identity (item 5); emotional aspects of illness representations are assessed by two items addressing concerns (item 6) and emotional impacts (item 8). Illness comprehensibility is assessed by one item: coherence (item 7). However, the causal dimension, which is an open-ended question (item 9), involves patients identifying their three most important perceived causes of T2DM. Because of its qualitative nature, the causal dimension of representation was excluded from the present study. All items are scored on an 11-point Likert scale (range: 0~10), with higher scores indicating that individual perceptions were more threatened by the illness. In the calculation, scores of items 3, 4, and 7 are reversed and added to those of items 1, 2, 5, 6, and 8. The test–retest reliability correlation coefficients of the original English BIPQ ranged from 0.48 to 0.70. Overall, the B-IPQ offers a quick evaluation of disease perceptions that is useful in large-scale trials of chronic diseases with repeated-measures study designs [10,12].

#### 2.4.2. Assessments of Comparative Measures of Depression, Anxiety, and Stress

Physiological disorders were measured by the 21-item Depression, Anxiety, and Stress Symptoms (DASS-21) questionnaire. The possible ranges of scores for each subdomain were as follows: DASS-21 scores of ≥10 for those with depression, ≥8 for those with anxiety, and ≥15 for those with stress, with a higher score representing a higher level of stress, anxiety, and depression symptoms [27]. The Bahasa version of depression (Cronbach’s α = 0.87), anxiety (Cronbach’s α = 0.85), and stress subdomains (Cronbach’s α = 0.72) demonstrated acceptable reliabilities [28].

#### 2.4.3. Assessments of Comparative Measures of QoL

The most widely used 36-item Short Form Survey (SF-36) to assess QoL is available on the RAND Corporation’s website [29]. Compared to diabetes-specific assessments, the SF-36 is able to measure an individual’s mental component score (MCS) and physical component score (PCS) and estimate their overall QoL. Moreover, the SF-36 has often been used to determine the effects of diabetes and predict responses to various diabetes treatments [27,29,30]. The initial SF-36 encompassed eight domains, role emotional (RE), bodily pain (BP), role physical (RP), social functioning (SF), vitality (VT), physical functioning (PF), mental health (MH), and general health (GH). Interestingly, these eight dimensions were simplified into three major domains, namely total QoL, MCS, and PCS. The total QoL score is the sum of the MCS and PCS. MCS is the sum of the scores of VT, SF, RE, and MH. PCS is the sum of the scores of GH, BP, RP, and PF. Thus, MCS, PCS, and total QoL range from 0 to 100, with a higher score indicating good QoL [27]. Notably, the SF-36 Indonesian version had a strong internal coherence (Cronbach’s alpha of 0.79) and test–retest reliability (*r* = 0.626~1) [31]. 

#### 2.4.4. Assessments of Comparative Measures of Fasting Blood Glucose (FBG)

After 12 h of fasting, all respondents were invited to assess blood glucose control using a fasting blood sample (5 mL) that was withdrawn from an antecubital vein. The blood was mixed with dipotassium ethylenediaminetetraacetic acid (1.5 to 2.2 mg/mL) and analyzed on an XP-100 automated hematology cell counter (Sysmex, Kobe, Japan) [32].

### 2.5. Statistical Analyses

To investigate the B-IPQ, measured values were determined, such as the minimum, maximum, mean, standard deviation (SD), skewness, and kurtosis. Skewness analyzes the propensity of data to be displaced from the baseline. A skewness of between −1 and 1 indicates acceptable skewness, and a skewness of >1 indicates a strongly skewed distribution [33]. Kurtosis is an indicator of the degree of vertical spread in a distribution around a mean, and a kurtosis of <2.5 times the standard error indicates a normal distribution [34]. The internal consistency reliability was calculated by Cronbach’s alpha test; a value ≥0.7 indicates sufficient internal consistency [25], and a corrected-item total correlation of >0.30 is considered adequate [35]. To calculate the test–retest reliability of the Bahasa version of the B-IPQ, the interclass correlation coefficient (ICC) was used, and a value of ≥0.75 suggests stability and reasonable test–retest reliability [36]. 

Exploratory factor analysis (EFA) and confirmatory factor analysis were used to assess construct validity (CFA). We split the data into two different data sets for our research. The first stage involved conducting EFA with varimax rotation on the first half of the independent sample for the major analysis factor. EFA was used to perform an initial analysis of the CFA model on the first half of the independent sample. For EFA, the adequacy of the data was verified using the Kaiser–Meyer–Olkin (KMO) test with >0.6 deemed appropriate, and *p* < 0.001 was deemed an appropriate result for Bartlett’s test of sphericity [37]. A factor loading of ≥0.35 was used to identify items as adequately reflecting the factor [22]. In order to establish the quality assumptions for adjustment of the assessment model, a CFA was undertaken to confirm the model–data fit. To determine an acceptable fit of the model to the data, the chi-squared (χ^2^) test value divided by the degrees of freedom (*df*) should be ≤3, the root mean square error of approximation (RMSEA) value should be ≤0.10 with the 90% confidence interval (CI), the standardized root mean square residual (SRMR) value should be ≤0.08, the comparative fit index (CFI) value should be ≥0.85, the Tucker–Lewis index (TLI) value should be ≥0.08, and the Akaike information criterion (AIC) is assessed as the smaller the better [23,24]. The χ^2^ fit index evaluates how well a hypothesized model fits data from a set of measuring items. Multivariate normality of data, acceptable sample size, no structured incomplete data, and sufficient model specification are all requirements of the χ^2^ model fit index [23,24]. Additionally, the convergent validity was examined by measuring the average variance extracted (AVE) and composite reliability (CR). Values of the AVE of >0.50 [23] and CR of >0.70 [38] were identified as appropriate. Moreover, Pearson’s correlation coefficients with *p* value of <0.05 were used to explore the convergent and divergent validities [11,23]. Data analyses were conducted using the Statistical Package for the Social Sciences (SPSS) Version 21 (SPSS, Chicago, IL, USA) and AMOS Version 25 (SPSS, Chicago, IL, USA).

## 3. Results

### 3.1. Sample Characteristics of Participants

Characteristics of participants are summarized in Table 2. The mean age was 55.44 years with a standard deviation (SD) of 6.92 years, and the average FBG level was 304.49 (27.04) mg/dL. Mean (SD) DASS-21 stress, anxiety, and depression scores were 13.65 (2.99), 8.02 (0.73), and 13.61 (5.04), respectively. Mean (SD) SF-36 total QoL score, MCS, and PCS were 47.48 (8.48), 49.31 (10.38), and 45.66 (9.47), respectively. The majority of participants were female, unmarried, and had an educational ISCED level of <3, a body mass index of <25 kg/m^2^, a nonsmoking status, and low physical activity (Table 2). 

The average CVI of the eight-item B-IPQ was 0.95, and another key finding of the test–retest reliability was that ICC results ranged from 0.82 to 0.90. The average score of the eight-item B-IPQ and the domain scores are shown in Table 3. The highest mean score was 7.25 (SD: 2.69) for illness concerns, while the lowest mean score was 5.64 (SD: 2.53) for treatment control. The skewness score for all B-IPQ items ranged from −0.76 to 1.26, and the kurtosis score ranged from −0.96 to 1.34 (Table 3).

### 3.2. Construct Validity 

#### 3.2.1. EFA of the B-IPQ

The principal axis factoring analysis indicated that the rotated factor loading value of the B-IPQ scale for all items was >0.35 with KMO was 0.80. The cognitive illness representation factors consisted of four items (consequences, timeline, personal control, and identity), and emotional representations involved three items (illness concerns, emotional, and coherency; Table 4). The reliability of the internal consistency with the item-total correlation coefficient score was 0.40 to 0.78 for each item. The Cronbach’s alpha total of the Indonesian version of the B-IPQ was 0.74, and the Cronbach’s alpha values of the domains were 0.84 for cognitive illness factor and 0.81 for emotional factor.

#### 3.2.2. CFA of the B-IPQ

The CFA conducted on the two-factor model was carried out with results χ^2^ = 43.26, *df* = 19, *p* < 0.001, SRMR = 0.06, RMSEA = 0.09, CFI = 0.96, TLI = 0.94, NFI = 0.93, GFI = 0.93, and AGFI = 0.87, which indicated the goodness of fit of the model structure. The goodness of fit for the model structure in this study is illustrated in Figure 1.

Interestingly, this study found that scores of CR were 0.85 and 0.81 for cognitive illness representations and emotional representations, respectively. Moreover, the two constructs of AVE values were 0.54 and 0.61 respectively, which showed favorable construct validity. 

A correlation analysis of the eight-item B-IPQ with depression, stress, anxiety, FBG, and all QoL domains was also performed to test the convergent and divergent validities of the instrument (Table 5). The overall B-IPQ was significantly positively correlated with cognitive illness representations (r = 0.90, *p* < 0.01), emotional representations (r = 0.84, *p* < 0.01), depression (r = 0.59, *p* < 0.01), stress (r = 0.51, *p* < 0.01), and anxiety (r = 0.55, *p* < 0.01), as well as FBG (r = 0.26, *p* < 0.01), and was significantly negatively correlated with total QoL (r = −0.59, *p* < 0.01), PCS (r = −0.44, *p* < 0.01), and MCS (r = −0.56, *p* < 0.01).

## 4. Discussion

To the best of our knowledge, this is the first study to examine patients with T2DM in Indonesia using multidimensional exploratory and confirmatory factor analyses and convergent and divergent validities. We highlighted the CR and AVE, which showed favorable key findings of construct validity. Moreover, participants were recruited from community-based area clinics that utilized stratified multistage clustering. The findings also demonstrated that the B-IPQ in Indonesians with T2DM is easy to use, requires less than 5 min for patients to complete, and takes only a few minutes to score. Therefore, it seems justified to conclude that the B-IPQ is a straightforward and simple survey.

In the present study, the overall Cronbach’s alpha value and those for the cognitive illness representation and emotional representation domains were adequate. This result is in line with previous research in Poland which showed that the internal consistency of the B-IPQ measured with Cronbach’s alpha of 0.74 was satisfactory for participants with myocardial infarction, neurological problems, rheumatoid arthritis, and lupus [15]. Moreover, an overall Cronbach’s alpha value of 0.78 was found in Ghana among T2DM patients [16], and Cronbach’s alpha of 0.75 was established in Korea with tuberculosis patients [17], The highlighted, limited studies that investigated the psychometric properties of the B-IPQ exhibited reliability using two-factor loading [11,35]. A previous study among Chinese with breast cancer revealed that Cronbach’s alpha values of overall, cognitive illness representations, and emotional representations were 0.78, 0.65, and 0.82, respectively [11]. Cronbach’s alpha values of overall, cognitive illness representations, and emotional representations were 0.85, 0.80, and 0.83, respectively, in Turkish cancer patients [35]. Moreover, the Cronbach’s alpha values of overall, cognitive illness representations, and emotional representations were 0.57, 0.68, and 0.83, respectively, in Malaysian cancer patients [39]. Furthermore, the B-IPQ seems acceptable for use with different disorders and across various cultures. In addition, the two-factor structure of the Bahasa version of the B-IPQ had acceptable internal consistency estimates with values of Cronbach’s alpha of >0.70, which indicates that the scale has solid reliability and internal consistency. Another key finding of the reliability was that ICC results ranged from 0.82 to 0.90, and the cognitive illness representation had the highest score compared to the emotional domain; these ICC values were higher than values in Malaysia [13], Korea [17], and Vietnam [40], for which score ranges were 0.53~0.78, 0.73~0.86, and 0.44~0.85, respectively. More generally, based on our ICC score and item-total correlation coefficient, we demonstrated that the Bahasa Indonesian version of the B-IPQ is a reliable and stable multidimensional instrument for measuring illness representations, including cognitive illness and emotional representations in Indonesians with T2DM.

Our results further showed that the EFA extracted two domains of the B-IPQ from the eight items in the questionnaire. The factor analysis results in this study differed from the original version. Our EFA results were consistent with Leventhal’s common sense model, which encompasses both emotional and cognitive dimensions of illness representations [11,35], that is, two loading factors of cognitive illness representations and emotional representations. Moreover, research conducted in two countries obtained similar factor loadings, but with different items. For example, the Turkish version recognized two-factor loading with seven items, excepting timeline [35], and the traditional Chinese version identified a two-factor loading with seven items, excepting illness coherence [11]. Based on the values of our factor loading of ≥0.35 used to identify items, it was determined to adequately and acceptably reflect the factor. Consequently, we considered maintaining the consequences item and illness coherence item to be more practicable for clinical decision-making and thus also beneficial to allow for intercultural proportionality. Our models also indicated the optimal goodness of fit, and our goodness of fit was close to prior research [35], which also reported RMSE (0.07), SRMR (0.06), CFI (0.97), and χ^2^/*df* (1.72) values. Additionally, taken together, the two-factor, eight-item solution retained in this study could reflect cultural viewpoints about illness perceptions among Indonesian individuals with T2DM.

The B-IPQ had moderate construct validity as the correlation coefficients between the clinical and patient-reported outcome measures demonstrated predicted values. This finding is congruent with former studies that documented well-known correlations of blood glucose with depression, anxiety, and stress in determining the convergent validity with the B-IPQ measurements. It appeared that individuals with T2DM and high levels of depression, anxiety, and stress tended to have high scores on cognitive and emotional aspects of their illness [11]. Our findings also revealed a meaningful correlation pattern in B-IPQ factors. Among them, high scores of cognitive and emotional factors were moderately correlated with high depression and anxiety scores [11]. According to previous research, depression, stress, anxiety, and blood glucose were proven to be easier to use, and the convergence of the B-IPQ was validated [11,13], which indicated a poor psychological level and low-level management of T2DM. Further evaluations of the convergent validity and its feasibility in T2DM populations in Indonesia are warranted.

Additionally, we explored correlations between the SF-36 QoL and Bahasa B-IPQ to clearly indicate the divergent validity. As hypothesized, we found moderately negative correlations of the Bahasa B-IPQ total score and cognitive and emotional illness representation domains with the SF-36 MCS, PCS, and total QoL score. Compared to diabetes-specific measurements, the SF-36 was able to assess MCSs and PCSs separately as well as calculating an overall QoL score. Moreover, it has been widely used in diabetes research, and researchers indicated that it is a truly reasonable choice for diabetes research, with relation to the inclusion of both mental and physical principles in evaluating the validity and responsiveness of SF-36 tools specifically related to T2DM [27,30]. Our findings are in line with the original B-IPQ [10] that specifically used the MCS-SF36 to explore the divergent validity and showed negative correlations with several items of the B-IPQ, such as cognitive domains (consequences and identity items) and emotional domains (concern and emotional response items). Moreover, the Persian B-IPQ total score was significantly negatively correlated with SF-36 PCS and MCS, which also confirmed the divergent validity [41,42]. Consequently, it is important to note that inverse correlations of the cognitive and emotional illness representations with the total QoL, PCS, and MCS will improve the quality of the content validity of the B-IPQ.

Among the limitations of the present study, we collected self-reported illness perception data using the B-IPQ, which may have caused a bias. Another concern is that the sample in this study was recruited from five community-based clinics in East Java, Indonesia, and enrolled respondents were female and of the Javanese ethnicity; consequently, the generalizability of our findings to all of Indonesia may be limited. Furthermore, we lacked information about diabetes medication, hemoglobin A1c, and comorbidities, which may affect the reported outcomes of illness perceptions. Further studies with comorbidities, diabetes medication, hemoglobin A1c levels, and related illness perceptions are also required to establish the diagnostic utility of the Bahasa version of the B-IPQ. Subsequent evaluation of the B-IPQ in this population should also consider the multilevel analyses that could furthermore clarify the effects multiethnic environments might have on illness perception and evaluate the specific identity subscale separately for rural and urban areas of participants.

## 5. Conclusions

In conclusion, our results indicate that the Bahasa version of the B-IPQ containing eight items with two-factor loading (cognitive and emotional illness representations) is an excellent-fitting multidimensional construct with convergent validity and internal consistency for measuring illness perceptions among individuals with T2DM. Moreover, these findings are significant since the questionnaire is brief and straightforward to perform in both academic and clinical settings.

## Figures and Tables

**Figure 1 ijerph-18-09601-f001:**
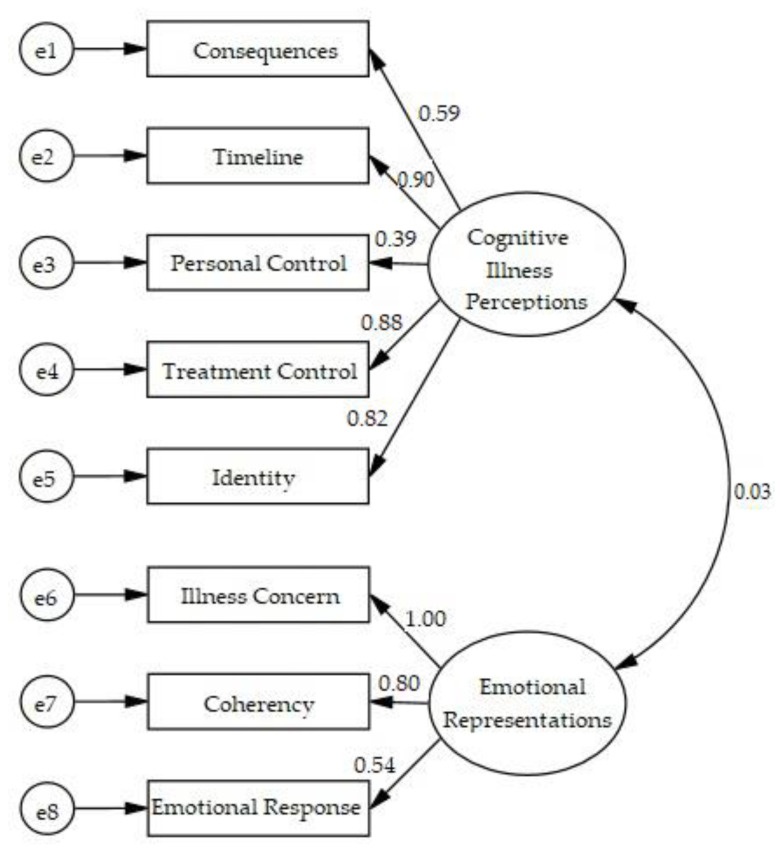
Standardized final models of the Brief Illness Perception Questionnaire among Indonesians with type 2 diabetes mellitus.

**Table 1 ijerph-18-09601-t001:** Psychometric analytical process of the Brief Illness Perception Questionnaire.

Description	Method	Result
Step 1: Translation process and content validity of the B-IPQ		
Forward and back translation	Conducted according to guidelines for cross-cultural adaption of self-reported measures [21]	A final version was accepted
2. Content validity	Expert review (validated by four expert reviews)	Mean CVI for the total scale was 0.95
Step 2: Psychometric analysis of the B-IPQ		
Validity		
Construct validity	EFA (factor loading ≥ 0.35) [22]	EFA (factor loadings were a minimum of 0.53 and maximum of 0.95)
	CFA (χ^2^/*df* ≤ 3.0, CFI and TLI ≥ 0.9, RMSEA ≤ 0.10, and AIC the smaller the better) [23,24]	The final factor model of the B-IPQ comprised eight items with two-factor loading and was confirmed with the following model fit indices: χ^2^/*df* (2.27), *p* < 0.001; CFI (0.96); TLI (0.94); RMSEA (0.09); and AIC (80.22)
b. Convergent and divergent validity	Pearson’s correlation [11,23]	Significantly correlated with the total QoL SF-36 (−0.59), subscales PCS (−0.44), and MCS (−0.56), depression (0.59), stress (0.51), anxiety (0.55), and FBG (0.26), all *p* < 0.01
2. Reliability		
Internal consistency	Cronbach’s α ≥ 0.7 [25]	Total scale and cognitive factor and emotional factor subscales were 0.74, 0.85, and 0.81, respectively

Notes: CVI, content validity index; EFA, exploratory factor analysis; CFA, confirmatory factor analysis; RMSEA, root mean square error of approximation; CFI, comparative fit index; TLI, Tucker–Lewis index; AIC, Akaike information criterion; *df*, degrees of freedom; QoL, quality of life; PCS, physical component score; MCS, mental component score; FBG, fasting blood glucose.

**Table 2 ijerph-18-09601-t002:** Sociodemographic and health-related status of participants.

Variable	Construct Validity (*n* = 294)			Test–Retest Reliability (*n* = 30)		
	Mean ± SD	*n*	%	Mean ± SD	*n*	%
Age (years)	55.44 ± 6.92			54.37 ± 5.74	30	100
Gender						
Female		237	80.6		27	90
Male		57	19.4		3	10
Marital status						
Not married		159	54.1		22	73.3
Married		135	45.9		8	26.7
Education						
ISCED < 3		169	57.5		23	76.7
ISCED ≥ 3		125	42.5		7	23.3
BMI (kg/m^2^)						
≥25		116	39.5		12	40
<25		178	60.5		18	60
Physical activity (MET-h/week)						
<7.5		273	92.9		29	96.7
≥7.5		21	7.1		1	3.3
Smoking						
Active smoker		33	11.2		3	10
Nonsmoker		261	88.8		27	90
FBG (mg/dL)	304.49 ± 27.04			303.27 ± 63.03		
DASS-21 stress	13.65 ± 2.99			16.0 ± 1.76		
DASS-21 anxiety	8.02 ± 0.73			7.87 ± 0.43		
DASS-21 depression	13.61 ± 5.04			14.0 ± 4.14		
SF-36 total QoL	47.48 ± 8.48			43.70 ± 8.73		
SF-36 MCS	49.31 ± 10.38			43.66 ± 10.92		
SF-36 PCS	45.66 ± 9.47			43.75 ± 7.47		

Notes: BMI, body mass index; FBG, fasting blood glucose; DASS-21, 21-item Depression, Anxiety, and Stress Scale; ISCED, International Standard Classification of Education; MCS, mental component score; MET, metabolic equivalent of task; PCS, physical component score; SD, standard deviation; SF-36, 36-item Short Form Health Survey; QoL, quality of life.

**Table 3 ijerph-18-09601-t003:** Average scores of the Bahasa version of the Brief Illness Perception Questionnaire according to items and domains (*n* = 294).

Item	Min	Max	Mean	SD	Skewness	Kurtosis
Cognitive illness						
Consequences	0.00	10	5.75	2.96	0.55	1.34
Timeline	0.00	10	5.80	3.06	−0.44	0.25
Personal control	0.00	10	6.14	3.12	−0.60	0.04
Treatment control	0.00	10	5.64	2.53	−0.38	0.15
Identity	0.00	10	5.77	2.69	0.61	−0.96
Emotional						
Illness concern	0.00	10	7.25	2.69	1.19	0.02
Emotional	0.00	10	7.16	2.87	1.26	0.10
Coherency	0.00	10	6.15	2.89	−0.76	−0.60

Notes: Min, minimum; Max, maximum; SD, standard deviation.

**Table 4 ijerph-18-09601-t004:** Factor structure of items of the Brief Illness Perception Questionnaire conforming to extracted factors after principal axis factoring analysis (*n* = 147).

Item no.	Item	Factor Loading	
		Cognitive Illness	Emotional
1	Consequences	0.64	−0.02
2	Timeline	0.85	−0.25
3	Personal control	0.45	0.11
4	Treatment control	0.85	−0.14
5	Identity	0.81	−0.17
6	Illness concern	0.22	**0.95**
7	Emotional	0.19	**0.80**
8	Coherency	0.11	**0.53**

Note: The highest value of factor loading is indicated in bold.

**Table 5 ijerph-18-09601-t005:** Correlations of the total score and subscales of the Brief Illness Perception Questionnaire with depression, stress, anxiety, and quality of life (*n* = 294).

Variable	B-IPQ	Cognitive Illness	Emotional
Cognitive illness representations	0.90 **		
Emotional representations	0.84 **	0.52 **	
DASS-21 depression	0.59 **	0.59 **	0.41 **
DASS-21 stress	0.51 **	0.42 **	0.48 **
DASS-21 anxiety	0.55 **	0.49 **	0.48 **
FBG (mg/dL)	0.26 **	0.24 **	0.23 **
SF-36 QoL	−0.59 **	−0.56 **	−0.46 **
SF-36 PCS	−0.44 **	−0.36 **	−0.42 **
SF-36 MCS	−0.56 **	−0.58 **	−0.37 **

Notes: ** *p* < 0.01. DASS-21, 21-item Depression, Anxiety, and Stress Scale; FBG, fasting blood glucose; SF-36, Short Form Survey; QoL, quality of life; PCS, physical component score; MCS, mental component score.

## Data Availability

The data presented in this study are available on request from the corresponding author.

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
