# Peer review of "Psychometric Testing of the Bahasa Version of the Brief Illness Perception Questionnaire among Indonesians with Type 2 Diabetes Mellitus"

_ijerph, 2021, doi:10.3390/ijerph18189601_

Round 1

Reviewer 1 Report

This contribution is very interesting. Topic is actual for this area and results are useful for practice. 
In introduction authors speak about this problematic in context (complex description). 
In methods I think that it can be useful to more describe chapter "2.3 Procedure" in view of another researches and recommendation about process of psychometric testing. 
Presented data are clear. Describe is detailed.
Discussion is adequate and it is supported by results and compare with literature. 

Author Response

RE- Manuscript ID: ijerph-1366819-Version 1

Response to Reviewer 1 Comments

Dear Reviewer 1,

Thank you for considering our manuscript and for the valuable suggestions, also the opportunity to resubmit a revised manuscript, which help us to improve the article. We carefully revised the manuscript in accordance with your comments. The revised sections of the manuscript are marked with blue color. Our point-by-point responses to the comments are as follows. We very much hope the revised manuscript is accepted for publication in International Journal of Environmental Research and Public Health. Thank you very much for your consideration.

Comments from Reviewer 1:

Point 1. This contribution is very interesting. Topic is actual for this area and results are useful for practice.

Response 1: Thank you very much.

Point 2. In introduction authors speak about this problematic in context (complex description).

Response 2: Thank you very much.

Point 3. In methods I think that it can be useful to more describe chapter "2.3 Procedure" in view of another researches and recommendation about process of psychometric testing.

Response 3: Thank you for your valuable suggestion. In order to make data to be useful and better presented, we add and reorganized the chapter “2.3 Procedure” based on the reviewer’s suggestion, as below; (please see line 128-143 on page 3).

“The Bahasa Indonesia version of the B-IPQ was assessed for translation process and psychometric analytical procedures including the following (1) we translated the English version of the B-IPQ into Bahasa Indonesia by a professional bilingual translator following recommended guidelines for cross-cultural adaptation assessment [21]. In brief. Indonesian-speaking academics were first contacted to review the translated version for grammatical accuracy and clarity. Thus, four independent bilingual translators completed the back translation of the Bahasa edition into English. Moreover, the final Bahasa version was obtained after comparing the original and back-translated questionnaires. Translators were instructed to use simple sentences and avoid metaphors, colloquial terminology, and hypothetical statements; (2) a pilot test of 10 Indonesians with T2DM was evaluated to clarify the wording and ease of understanding of the final version; (3) we reviewed cognitive debriefing results and the finalized version (content validity index (CVI) for the total scale = .95 with four expert reviews); (4) Finally, we conducted a psychometric analysis of the internal consistency, construct, convergent and divergent validities and summarized the analytical proses and results in table 1.”

Point 4. Presented data are clear. Describe is detailed.

Response 4: Thank you very much.

Point 5. Discussion is adequate and it is supported by results and compare with literature. 

Response 5: Thank you very much.

Reviewer 2 Report

Although the topic of the article is not novel, as the authors themselves point out in the introduction there are several versions of the instrument in different populations, It is interesting to see their approach in its abbreviated and translated version for the Indonesian population. 

Regarding the format of decimal numbers, following the APA regulation I suggest to eliminate the zero in alpha values, p values or adjustment index, for example. Similarly, attempt to remove the horizontal inner lines from the tables.

Author Response

RE- Manuscript ID: ijerph-1366819-Version 1

Response to Reviewer 2 Comments

Dear Reviewer 2,

Thank you for considering our manuscript and for the valuable suggestions, also the opportunity to resubmit a revised manuscript, which help us to improve the article. We carefully revised the manuscript in accordance with your comments. The revised sections of the manuscript are marked with blue color. Our point-by-point responses to the comments are as follows. We very much hope the revised manuscript is accepted for publication in International Journal of Environmental Research and Public Health. Thank you very much for your consideration.

Comments from Reviewer 2:

Point 1. Although the topic of the article is not novel, as the authors themselves point out in the introduction there are several versions of the instrument in different populations, It is interesting to see their approach in its abbreviated and translated version for the Indonesian population.

Response 1: Thank you very much.

Point 2. Regarding the format of decimal numbers, following the APA regulation I suggest to eliminate the zero in alpha values, p values or adjustment index, for example. Similarly, attempt to remove the horizontal inner lines from the tables.

Response 2: Thank you for your valuable suggestion. In order to make data to be better presented following the APA regulation, we eliminated the zero in alpha values, p values or adjustment index, as well as the horizontal inner lines from the tables based on the reviewer’s suggestion.

Reviewer 3 Report

  1. Summary of the research overall impression:

This study aimed to psychometrically the 8-item 27 B-IPQ among Indonesians with type 2 diabetes mellitus (T2DM).

Data were included 294 patients 28 with T2DM with stratified multistage.

I think the manuscript is really interesting, but, please:

  • The authors should improve the most important strengths of the study.

  • Abstract: please: insert a clear conclusion. Please focus the abstract on your study and your results.
  • Statistical analyses: IBM SPSS Statistics? Version? P value?
  • Participants : please, change N to n / N=population / n=sample (see results and tables).
  • I suggest improve more recent studies in the discussion session. See, for example, references: 3, 32, 40.
  • The authors should improve suggestions for future research and another studies.
  • The data should be provided to the Journal as part of the manuscript or its supporting information, or deposited to a public repository. For example, in addition to summary statistics, the data points behind means, medians and variance measures should be available. If there are restrictions on publicly sharing data—e.g. participant privacy or use of data from a third party—those must be specified.

  1. Additional comments, including concerns about dual publication, research ethics, or publication ethics:

-  Is the manuscript technically sound, and do the data support the conclusions?

Yes

- The manuscript must describe a technically sound piece of scientific research with data that supports the conclusions. Experiments must have been conducted rigorously, with appropriate controls, replication, and sample sizes. The conclusions must be drawn appropriately based on the data presented.

Yes

  • Has the statistical analysis been performed appropriately and rigorously?

 No

  • Have the authors made all data underlying the findings in their manuscript fully available?

No

  • Is the manuscript presented in an intelligible fashion and written in standard English?

Yes

- Any typographical or grammatical errors should be corrected at revision, so please note any specific errors in the paper.

Yours sincerely,

Author Response

RE- Manuscript ID: ijerph-1366819-Version 1

Response to Reviewer 3 Comments

Dear Reviewer 3,

Thank you for considering our manuscript and for the valuable suggestions, also the opportunity to resubmit a revised manuscript, which help us to improve the article. We carefully revised the manuscript in accordance with your comments. The revised sections of the manuscript are marked with blue color. Our point-by-point responses to the comments are as follows. We very much hope the revised manuscript is accepted for publication in International Journal of Environmental Research and Public Health. Thank you very much for your consideration.

Comments from Reviewer 3:

Summary of the research overall impression:

This study aimed to psychometrically the 8-item 27 B-IPQ among Indonesians with type 2 diabetes mellitus (T2DM). Data were included 294 patients 28 with T2DM with stratified multistage. I think the manuscript is really interesting, but, please:

Point 1. The authors should improve the most important strengths of the study.

Response 1: Thank you for your valuable comments. In this revised manuscript, we added some important strengths of the study in the first paragraph of the discussion section (please see line 296-301 on page 9).

“We highlighted the CR, and AVE showed the favorable key findings of the construct validity. Moreover, participants were recruited from community-based area clinics that utilized stratified multistage clustering. The findings also demonstrated that the B-IPQ in Indonesians with T2DM is easy to use, requires less than 5 minutes for patients to complete, and takes only a few minutes to score. Therefore, it seems justified to conclude that the B-IPQ is a straightforward and simple survey”.

Point 2. Abstract: please: insert a clear conclusion. Please focus the abstract on your study and your results.

Response 2: Thank you for your valuable comments and suggestions. In order to make sentences to be better presented, we reorganized the abstract based on the reviewer’s suggestion (please see line 38-40 on page 1).

“Findings provide empirical evidence that the Bahasa version of the B-IPQ showed adequate internal consistency, exploratory and confirmatory, thus, valid and reliable for illness perception assessments among Indonesians with T2DM”

Point 3. Statistical analyses: IBM SPSS Statistics? Version? P value?

Response 3: Thank you for your valuable comments. Let we clarify and revise this point to make it clearer and more precise based on the reviewer’s suggestion as follows in the statistical analyses section of our study

“Data analysis were conducted using the Statistical Pack- age for the Social Sciences (SPSS) Vers. 21 (SPSS, Chicago, IL, USA), and AMOS Vers. 25 (SPSS, Chicago, IL, USA). “(please see line 231-233 on page 6).

“Moreover, Pearson’s correlation coefficients with p value of <.05 were used to explore the convergent and divergent validities” (please see line 229-230 on page 6).

Point 4. Participants: please, change N to n / N=population / n=sample (see results and tables).

Response 4: Thank you for your valuable comments. We revised this point follow reviewer’s suggestion (please see table 2, 3, 4 and 5).

Point 5. I suggest improve more recent studies in the discussion session. See, for example, references: 3, 32, 40

Response 5: Thank you for your valuable comments. In this revised manuscript, we added two recent studies in the discussion section the above point of

 “Moreover, the Cronbach’s alpha value was .57, the cognitive illness representations was .68, and emotional representations was .83 in Malaysian cancer patients (39)”. (please see line 314-316 on page 9).

Reference number 39: Rajah, H.D.A.; Chie, Q.T.; Ahmad, M.; Leong, W.C.; Bhoo-Pathy, N.; Chan, C.M.H. Reliability and Validity of the Brief Illness Perception Questionnaire in Bahasa Malaysia for Patients with Cancer. Asian Pacific Journal of Cancer Prevention: APJCP 2021, 22, 2487-2492

 “Moreover, the Persian B-IPQ total score was significantly negatively correlated with SF-36 PCS and MCS which also confirmed the divergent validity (42)”. (please see line 371-373 on page 10).

Reference number 42: Seydi, M.; Akhbari, B.; Abad, S.K.G.; Jaberzadeh, S.; Saeedi, A.; Ashrafi, A.; Shakoorianfard, M.A. Psychometric properties of the Persian Version of the Brief Illness Questionnaire in Iranian with Non-specific Chronic Neck pain. Journal of Bodywork and Movement Therapies 2021

Point 6. The authors should improve suggestions for future research and another studies.

Response 6: Thank you for your valuable comments. In this revised manuscript, we added the above point of (please see line 384-388 on page 11).

“Subsequent evaluation of the B-IPQ in this population should also consider the multilevel analyses that could furthermore clarify the effects multiethnic environments might have on illness perception and needed to evaluate the specific identity subscale, separately for rural and urban areas of participants”

Point 7. The data should be provided to the Journal as part of the manuscript or its supporting information, or deposited to a public repository. For example, in addition to summary statistics, the data points behind means, medians and variance measures should be available. If there are restrictions on publicly sharing data—e.g. participant privacy or use of data from a third party—those must be specified.

Response 7: Thank you for your valuable comments. The data presented in this study are available on request from the corresponding author or the first author.

Point 8. Additional comments, including concerns about dual publication, research ethics, or publication ethics:

Response 8: Thank you very much. This manuscript is original and is not being considered for publication elsewhere. No conflict of interest exists with the submission of this manuscript, and all authors approved the manuscript for publication.

The study was conducted according to the guidelines of the Declaration of Helsinki, and approved by the Institutional Review Board of Institut Ilmu Kesehatan Strada Indonesia (IRB/009/KET-TPEP/X-2018) and conformed to the Declaration of Helsinki. All participants were allowed to participate in the research after they had received verbal and written information related to our research.

Point 9. Is the manuscript technically sound, and do the data support the conclusions? Yes

Response 9: Thank you very much.

Point 10. The manuscript must describe a technically sound piece of scientific research with data that supports the conclusions. Experiments must have been conducted rigorously, with appropriate controls, replication, and sample sizes. The conclusions must be drawn appropriately based on the data presented? Yes

Response 10: Thank you very much.

Point 11. Has the statistical analysis been performed appropriately and rigorously? No

Response 11: Thank you very much. Let we clarify and revise our statistical analysis to make it appropriate and rigorously based on the reviewer’s suggestion in the statistical analyses section of our study as follows;

“Data analysis were conducted using the Statistical Pack- age for the Social Sciences (SPSS) Vers. 21 (SPSS, Chicago, IL, USA), and AMOS Vers. 25 (SPSS, Chicago, IL, USA). “(please see line 224-225 on page 6).

 The statistical analysis we summarized in the abstract section:

“The validity analysis included the construct and convergent with significant person correlations. Cronbach’s alpha, composite reliability (CR), and average variance extracted (AVE) for reliability. Confirmatory and exploratory factor analyses (CFA and EFA) indicated a multidimensional structure, including cognitive with a five-item structure and emotional illness representation with a three-item structure, with an acceptable goodness-of-model fit” “(please see line 31-34 on page 1).

Also, we add more information and resentences related the specific analysis for EFA and CFA

“Exploratory factor analysis (EFA) and confirmatory factor analysis were used to assess construct validity (CFA). We split the data into two different data sets for our research. The first stage involved conducting EFA with varimax rotation on the first half of the independent sample for the major analysis factor. EFA was used to perform an initial analysis of the CFA model on the first half of the independent sample.” (please see line 209-213 on page 5).

“To determine an acceptable fit of the model to the data, a Chi-squared (X2) test divided by the degrees of freedom (df) should be ≤3, root mean square error of approximation (RMSEA) value should be ≤.10 with the 90% confidence interval (CI), the standardized root mean square residual (SRMR) value should be ≤.08, the comparative fit index (CFI) value should be ≥.85, Tucker Lewis Index (TLI) value should be ≥.08 and the Akaike information criterion (AIC) is assessed as the smaller, the better” (please see line 212-226 on page 5-6).

Point 12. Have the authors made all data underlying the findings in their manuscript fully available? No

Response 12: Thank you very much. In this manuscript, all data underlying the findings in our manuscript was fully available.  We add some information related our data to make it clear and rigorously based on the reviewer’s suggestion in the statistical analyses section of our study as follows;

“A skewness of between -1 and 1 indicates acceptable skewness, and a skewness of >1 indicates a strongly skewed distribution [33]. Kurtosis is an indicator of the degree of vertical spread in a distribution around a mean, and a kurtosis of <2.5 times the standard error indicates a normal distribution” (please see line 200-204 on page 5). “

The X2 fit index evaluates how well a hypothesized model fits data from a set of measuring items. Multivariate normality of data, acceptable sample size, no structured incomplete data, and sufficient model specification are all requirements of the X2 model fit index” (please see line 224-227 on page 6).

Point 13. Is the manuscript presented in an intelligible fashion and written in standard English? Yes

Response 13: Thank you very much.

Point 14. Any typographical or grammatical errors should be corrected at revision, so please note any specific errors in the paper.

Response 14: Thank you for your valuable suggestion. This manuscript was edited by Taipei Medical University Academic Editing.

Round 2

Reviewer 3 Report

Kind Regards,

GD
